# EXAFS Determination of Clay Minerals in Martian Meteorite Allan Hills 84001 and Its Implication for the Noachian Aqueous Environment

**Ryoichi Nakada [1],\*, Gaku Tanabe [2], Iori Kajitani [2,3], Tomohiro Usui [3,4], Masashi Shidare [2] and Tetsuya Yokoyama [2]**

[1] Kochi Institute for Core Sample Research, Japan Agency for Marine-Earth Science and Technology (JAMSTEC), 200 Monobe, Nankoku, Kochi 783-8501, Japan

[2] Department of Earth and Planetary Sciences, Tokyo Institute of Technology, 2-12-1 Ookayama, Meguro, Tokyo 152-8551, Japan; gaaaaaku7130@gmail.com (G.T.); kajitani@g.ecc.u-tokyo.ac.jp (I.K.); ms.b10.theta@gmail.com (M.S.); tetsuya.yoko@eps.sci.titech.ac.jp (T.Y.)

[3] Institute of Space and Astronautical Science, Japan Aerospace Exploration Agency, 3-1-1 Yoshinodai, Chuo, Sagamihara, Kanagawa 252-5210, Japan; usui.tomohiro@jaxa.jp

[4] Earth-Life Science Institute, Tokyo Institute of Technology, 2-12-1 Ookayama, Meguro, Tokyo 152-8551, Japan

\* Correspondence: nakadar@jamstec.go.jp; Tel.: +81-88-878-2275; Fax: +81-88-878-2192

**Abstract:** The aqueous environment of ancient Mars is of significant interest because of evidence suggesting the presence of a large body of liquid water on the surface at ~4 Ga, which differs significantly from the modern dry and oxic Martian environment. In this study, we examined the Fe-bearing minerals in the 4 Ga Martian meteorite, Alan Hills (ALH) 84001, to reveal the ancient aqueous environment present during the formation of this meteorite. Extended X-ray absorption fine structure (EXAFS) analysis was conducted to determine the Fe species in ALH carbonate and silica glass with a high spatial resolution (~1–2 μm). The μ-EXAFS analysis of ALH carbonate showed that the Fe species in the carbonate were dominated by a magnesite-siderite solid solution. Our analysis suggests the presence of smectite group clay in the carbonate, which is consistent with the results of previous thermochemical modeling. We also found serpentine in the silica glass, indicating the decrease of water after the formation of carbonate, at least locally. The possible allochthonous origin of the hematite in the carbonate suggests a patchy redox environment on the ancient Martian surface.

**Keywords:** Martian meteorite; ALH 84001; carbonate; clay mineral; EXAFS





## 1. Introduction

Geological and geomorphological evidence suggests the presence of a large amount of liquid water on the surface of early Mars [1–5]. Observations by both the orbiter and rover detected a variety of hydrous minerals and evaporites [6–9]. The global distribution of clay minerals suggests a long-term interaction between water and a basaltic crust during the Noachian and Hesperian periods [7,8,10]. An early study based on the compact reconnaissance imaging spectrometer for Mars (CRISM) categorized the types and distribution of clay minerals with co-occurring alteration phases into three different types and showed the dominance of Fe/Mg smectite irrespective of the type [11]. Smectites were also observed in crater-rim-derived rocks in the Gale crater, possibly formed by diagenesis in the early Hesperian [12,13]. Despite their ubiquity on the Martian surface, clay minerals have only been observed in small-scattered regions in a ~4.1 Ga Martian meteorite, Allan Hills (ALH) 84001 [14].

ALH 84001 is a unique Noachian igneous rock [15] containing secondary carbonate minerals and silica glass [16–18]. Carbonate minerals in ALH 84001 (hereafter referred to as ALH carbonates) can be divided into two groups, i.e., "rosette-type" spheroid-zoned concretions and massive ankeritic "slab-like" domains [19,20]. ALH carbonates are considered to have precipitated from a low-temperature aqueous alteration at the Martian near

surface at 4.04–3.90 Ga [21,22]. In addition to smectite-type clay, fine-grained or nanophase magnetite ($Fe_3O_4$) and Fe-sulfide have been observed in ALH carbonate using transmission electron microscopy (TEM) [23–25]. A study using secondary ion mass spectrometry (SIMS) showed that the ALH carbonate contained 0.4–1.1 wt.% $H_2O$ [26], which is much higher than that in the bulk contents (0.08 wt.% [27]). This suggests the presence of water in ALH carbonate as well as the occurrence of hydrous minerals such as clay minerals, goethite, and ferrihydrite. Model calculations indeed suggest the possible coprecipitation of goethite with carbonate [28,29], although at present, goethite has not been found in ALH carbonate. This inconsistency may be caused by the difficulty in identifying trace amounts of contaminants in the carbonate matrix. In the X-ray diffraction analysis of ALH carbonate, peaks other than carbonate minerals would not be discernible.

X-ray absorption fine structure (XAFS) spectroscopy is sensitive to the local structure of an element of interest (i.e., coordination environment), such that the method enables the distinction of minerals with different chemical structures. Because the X-ray energy where the absorption occurs only depends on the element, XAFS analysis is unaffected by the matrix composition unless the adsorption edge is located around a similar energy region (e.g., V K-edge and La $L_{III}$-edge). The identification of Fe-bearing minerals, including clay minerals from sedimentary rocks, was successfully achieved using this technique [30–32]. In addition, an X-ray beam can be focused at the micrometer scale such that a speciation analysis using XAFS can be applied with a high spatial resolution (μ-XAFS). XAFS analysis has been applied to Martian meteorites in the speciation analyses of Fe [33,34], S [35,36], V [37], and N [38]. These advantages enable us to determine the Fe-bearing minerals, including clay minerals preserved in the ALH carbonates measuring ~50–200 μm using Fe K-edge μ-XAFS. Although it may be possible that Fe-bearing carbonates such as siderite and ankerite dominate the Fe species in the ALH carbonate, the contribution from a small amount of other Fe species can be identified by the extended μ-XAFS (μ-EXAFS) spectrum.

In this study, a μ-EXAFS analysis of Fe in ALH carbonate was conducted to investigate Fe species and assess the ancient aqueous environment of Mars. We also conducted a μ-EXAFS analysis of Fe in silica glass around ALH 84001 carbonate. The comparison of Fe-bearing minerals between carbonate and silica glass enables the discussion of the evolution of a fluid after the formation of ALH carbonates because silica glass is, in a secondary phase, formed by another fluid after carbonate formation [39].

## 2. Sample and Method

### 2.1. Sample

A polished thin section of ALH 84001 (#81), allocated from the NASA Johnson Space Center in Houston, Texas, was used in this study. Back-scattered electron (BSE) and X-ray images of Ca, Mg, and Fe in the carbon-coated section were obtained using scanning electron microscopy and energy dispersive X-ray spectroscopy (SEM-EDS; S-3400N, Hitachi, Tokyo, Japan) at the Tokyo Institute of Technology. We selected four carbonate-rich regions (termed Areas 1–4, as shown in Figure 1) covering carbonates with various properties for the Fe K-edge μ-EXAFS analysis. Areas 1 and 3 are composed of rosette-type carbonate, whereas Areas 2 and 4 are classified as slab-like carbonates (Figure 2 and Table 1).

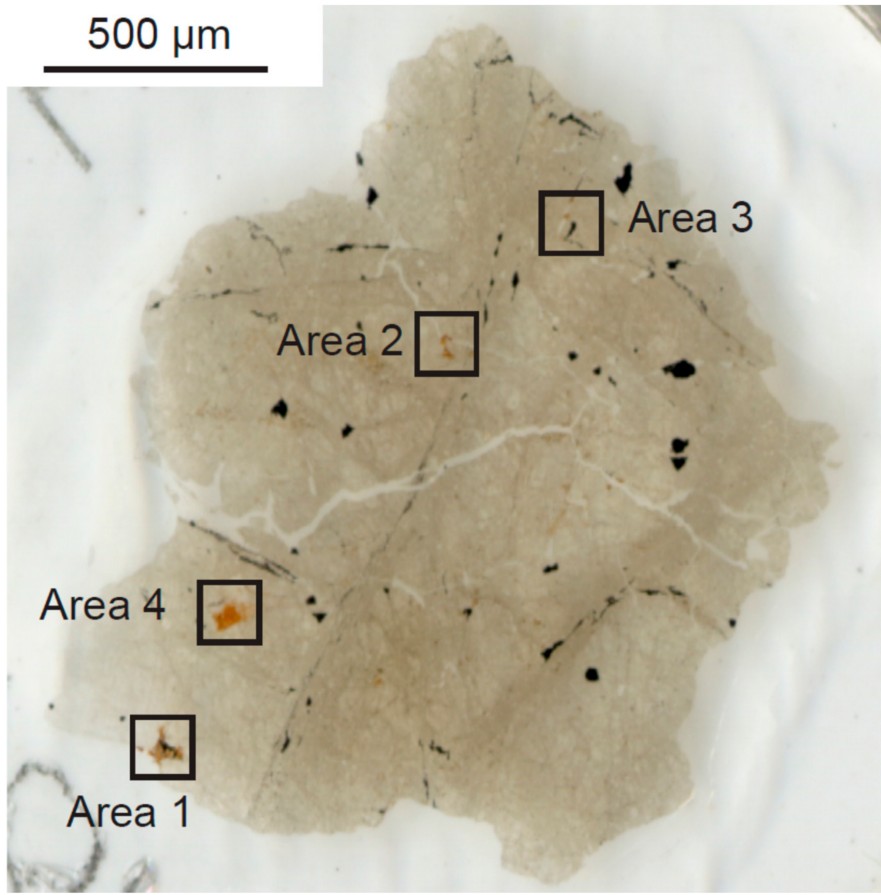

**Figure 1.** Photograph (plane-polarized light) of the thin section of Alan Hills (ALH) 84001 used in this study. Boxes identify the X-ray fluorescence (XRF) mapping areas shown in Figure 2.

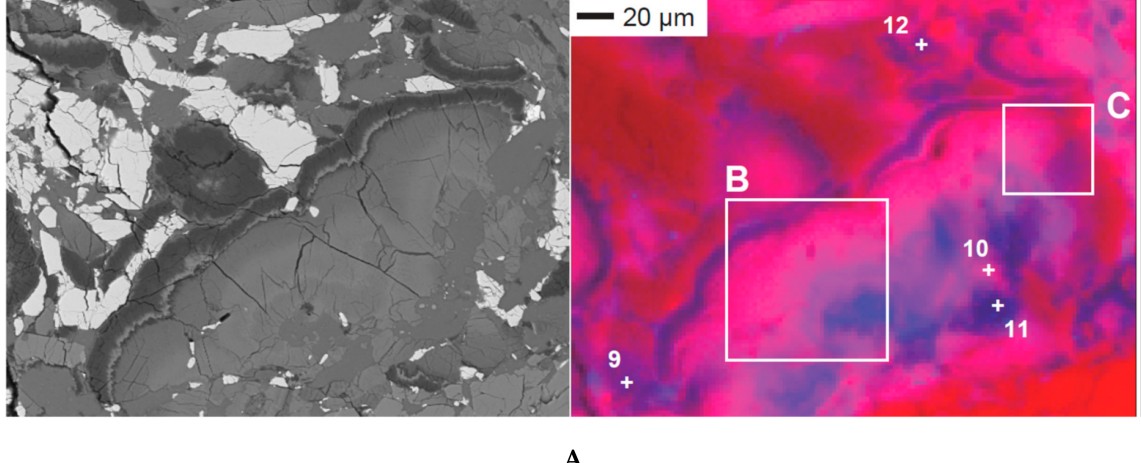

**A**

**Figure 2.** *Cont.*

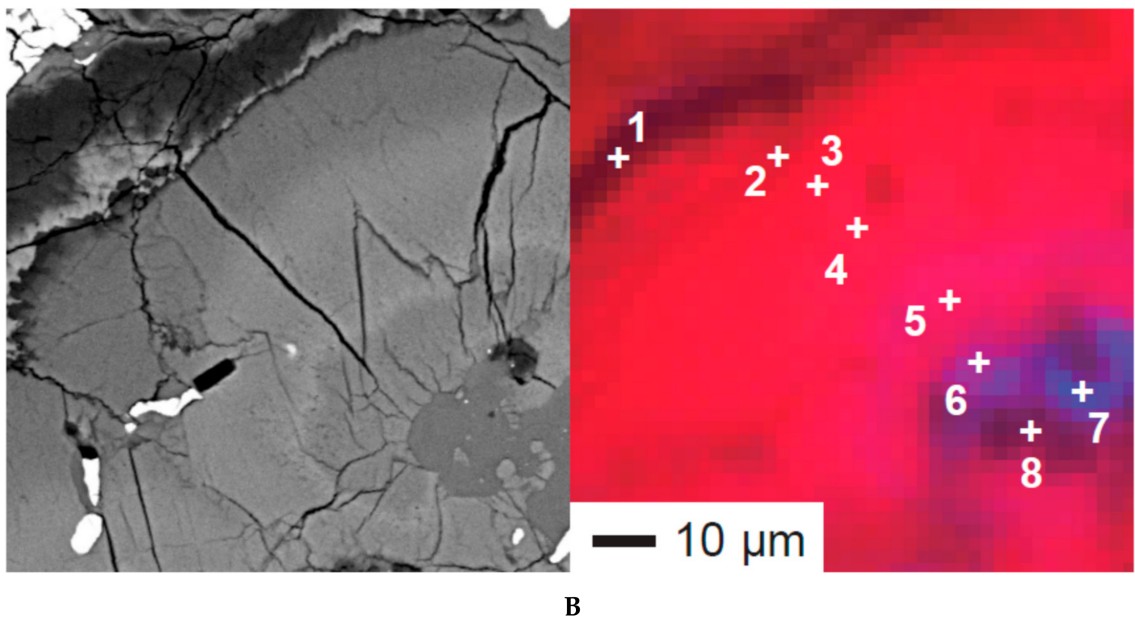

**B**

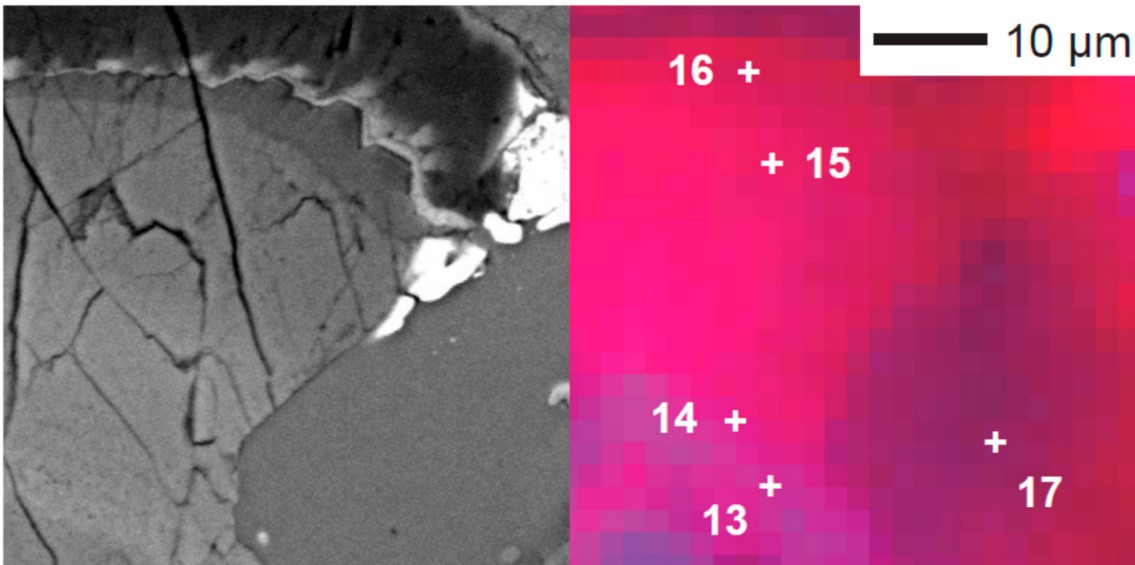

**C**

**Figure 2.** *Cont.*

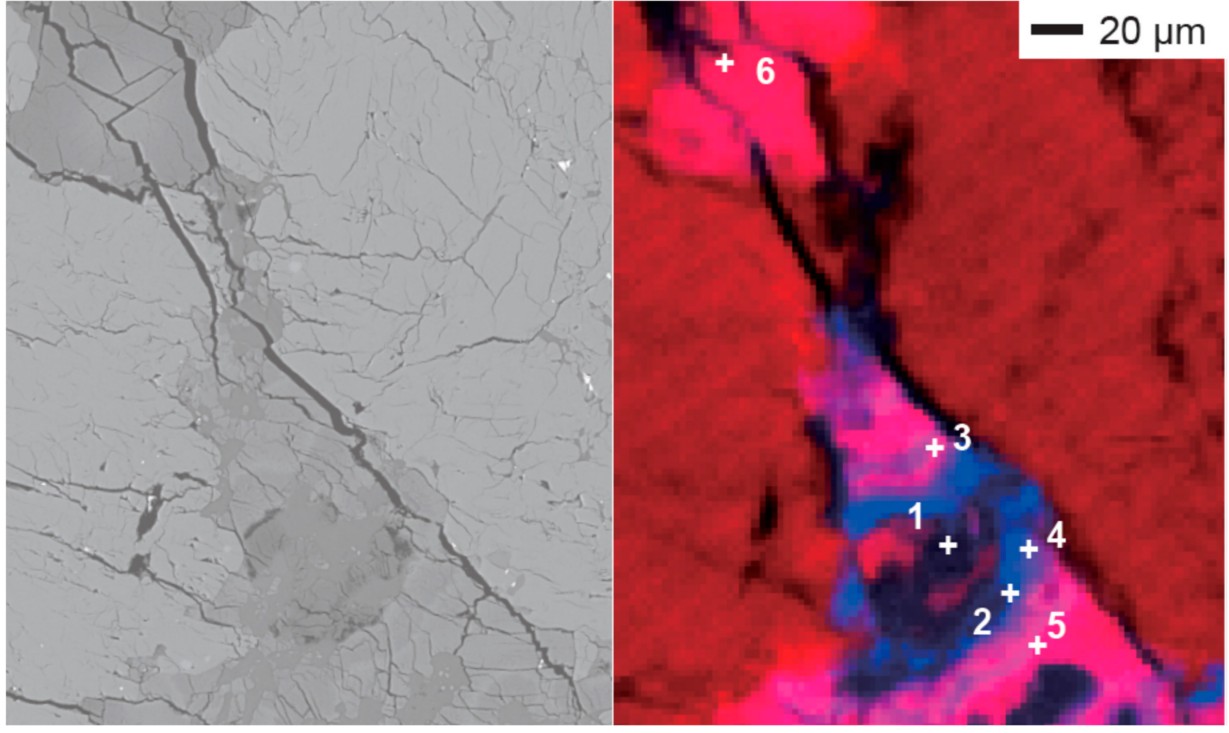

D

E

**Figure 2.** *Cont.*

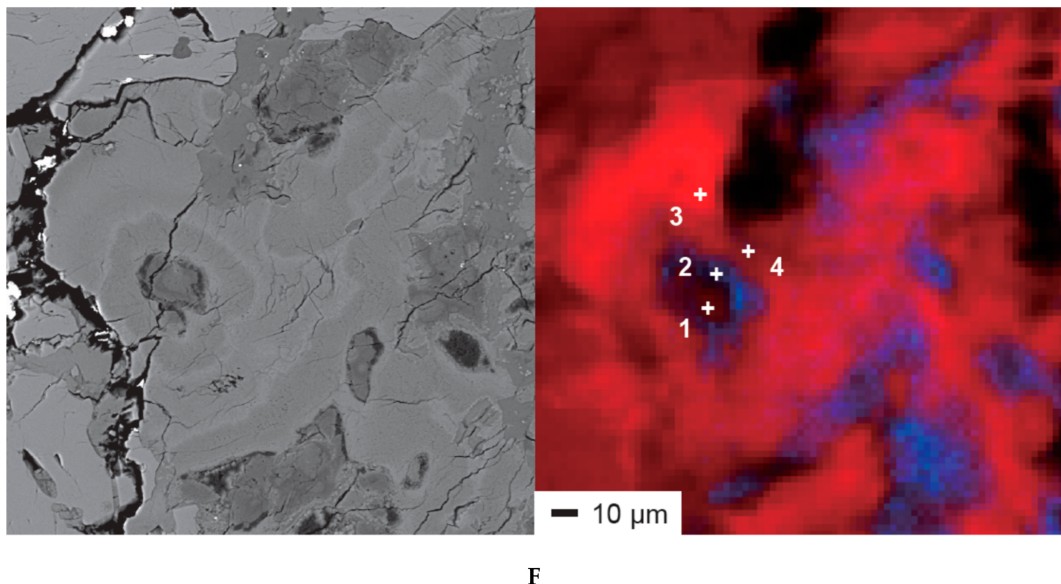

**F**

**Figure 2.** Backscattered electron (BSE) and synchrotron XRF mapping (Fe = red; Ca = blue) images. (**A**–**C**) Area 1, (**D**) Area 2, (**E**) Area 3, and (**F**) Area 4. The white crosses shown in the XRF mapping denote the μ-extended X-ray absorption fine structure (EXAFS) analytical spots.

**Table 1.** Description on the μ-EXAFS analytical spots.

| Analytical Spots | | Description |
|---|---|---|
| *Area 1* | spot 1 | Rim of a Mg-rich rosette carbonate |
| | spot 2 | Between Mg-rich rim and Fe-rich core of the rosette carbonate |
| | spot 3 | Fe-rich core of the rosette carbonate |
| | spot 4 | Fe-rich core of the rosette carbonate |
| | spot 5 | Fe-rich core of the rosette carbonate |
| | spot 6 | Ca-rich core of the rosette carbonate |
| | spot 7 | Ca-rich core of the rosette carbonate |
| | spot 8 | Silica glass located in the rosette carbonate |
| | spot 9 | Silica glass adjacent to the rosette carbonate |
| | spot 10 | Fe-rich core of the rosette carbonate |
| | spot 11 | Silica glass located in the rosette carbonate |
| | spot 12 | Silica glass |
| | spot 13 | Fe-rich core of the rosette carbonate |
| | spot 14 | Fe-rich core of the rosette carbonate |
| | spot 15 | Between Mg-rich rim and Fe-rich core of the rosette carbonate |
| | spot 16 | Rim of a Mg-rich rosette carbonate |
| | spot 17 | Silica glass located in the rosette carbonate |
| *Area 2* | spot 1 | Silica glass located in the slab carbonate |
| | spot 2 | Slab carbonate closely adjacent to the silica glass |
| | spot 3 | Fe-rich slab carbonate |
| | spot 4 | Ca-rich slab carbonate |
| | spot 5 | Slab carbonate with moderate Fe contents |
| | spot 6 | Slab carbonate with moderate Fe contents |
| *Area 3* | spot 1 | Silica glass |
| | spot 2 | Core of the rosette carbonate rich in Ca |
| | spot 3 | Core of the rosette carbonate rich in Fe |
| | spot 4 | Between Mg-rich rim and Fe-rich core of the rosette carbonate |
| *Area 4* | spot 1 | Slab carbonate with moderate Fe |
| | spot 2 | Slab carbonate with moderate Fe |
| | spot 3 | Fe-rich slab carbonate |
| | spot 4 | Slab carbonate with depleted Ca |

### 2.2. Iron K-Edge EXAFS

2.2.1. Reference Materials

The Fe K-edge (7111 eV) EXAFS spectra were measured at BL-12C of the Photon Factory (PF; Tsukuba, Japan). A white beam from a bending magnet was monochromatized using a Si(111) double-crystal monochromator. An Rh-coated bent cylinder mirror focused the X-ray beam to a final spot size of 0.6 mm (H) × 0.6 mm (V). A double flat mirror was inserted to reduce the third harmonic. Measurements were conducted at ambient pressure and temperature. Prior to the measurements, the X-ray energy was calibrated with the pre-edge peak maximum $Fe_2O_3$ (hematite) at 7111 eV, which enabled the comparison of the spectra obtained at different beamlines. The majority of the EXAFS spectra were acquired in transmission mode, whereas some samples were obtained in fluorescence mode using the Lytle detector.

Reference materials of carbonates, including ankerite, dolomite, magnesite, siderite, and sulfides such as pyrite and pyrrhotite were obtained from Hori Mineralogy, Japan. Although pure magnesite ($MgCO_3$) does not contain Fe, magnesite forms a solid solution with siderite ($FeCO_3$) such that a small amount of Fe could be incorporated as an impurity in the magnesite used in this study. Unfortunately, we could not obtain a reference of the calcite-siderite solid solution. Jarosite was obtained from the National Museum in Japan. Iron sulfate ($FeSO_4 \cdot H_2O$), hematite, and magnetite were obtained from FUJIFILM Wako Pure Chemical Co. (Osaka, Japan). Goethite and ferrihydrite were synthesized according to the procedure described in [40]. Clay minerals such as chlorite (CCa-2), illite (IMt-2), illite-smectite mixed layer (ISCz-1), kaolinite (KGa-1b), nontronite (NAu-1, NAu-2, NG-1), and montmorillonite (STx-1b and SWy-3) were obtained from the Source Clays Repository of the Clay Mineral Society, USA. A low-temperature type serpentine, composed of lizardite and chrysotile with a small amount of magnetite, was collected from Chiba, Japan [41]. A high-temperature type serpentine composed of antigorite with a small amount of magnetite was collected from Nagasaki, Japan [42]. The details of clay minerals are given in Table 2.

**Table 2.** Description on the reference clay minerals.

| Name | Mineral | Locality | Reference |
|---|---|---|---|
| CCa-2 | Ripidolite (chlorite) | California, USA | [43] |
| IMt-2 | Illite | Montana, USA | [43] |
| ISCz-1 | Illite-smectite mixedlayer (70/30 ordered) | Slovakia | [43] |
| KGa-1b | Kaolin (low-defect) | Georgia, USA | [43] |
| NAu-1 | Nontronite (Al-enriched) | South Australia | [43] |
| NAu-2 | Nontronite (Al-poor) | South Australia | [43] |
| NG-1 | Nontronite | Hohen Hagen, Germany | [43] |
| STx-1b | Ca-montmorillonite | Texas, USA | [43] |
| SWy-3 | Na-rich Montmorillonite | Wyoming, USA | [43] |
| Serpentine (L) | Lizardite and chrysotile | Chiba, Japan | [41] |
| Serpentine (H) | Antigorite | Nagasaki, Japan | [42] |

2.2.2. ALH Sample

The μ-EXAFS measurement is required for the ALH 84001 sample because the analytical area of carbonate and silica glass is approximately 200 × 200 μm², with a continuous change in composition. The Fe K-edge μ-EXAFS measurements were conducted at BL05XU of SPring-8 (Hyogo, Japan) under ambient temperature and pressure. The white beam from an undulator was monochromatized using a Si(111) double-crystal monochromator. The X-ray beam was focused using a K-B mirror to a final spot size of 1–2 μm (vertical) × 1–2 μm (horizontal) (beam size varying with the beamtime). Prior to the measurements, the X-ray energy was calibrated using the pre-edge peak maximum $Fe_2O_3$ (hematite) at 7111 eV measured in transmission mode. The EXAFS spectra of the ALH 84001 sample were acquired in fluorescence mode. In the latter mode, X-ray fluorescence (XRF) signals of the sample placed at 45° to the incident beam were obtained using a four-element

silicon drift detector (SDD) positioned 90° to the incident beam. The XRF mapping was obtained prior to the μ-EXAFS measurement scanned in 2 μm steps at 7200 eV to determine analytical spots with reference to the BSE. The EXAFS measurement of each analytical spot was scanned four times to obtain spectra with better statistics. Repeated EXAFS scans were identical, showing that any X-ray-induced alteration of the samples was unlikely during the measurements.

The EXAFS spectra were analyzed using the Athena software [44] to estimate the Fe species components and their mixture ratios. The energy unit was transformed from eV to $\text{Å}^{-1}$ to produce the EXAFS function $\chi(k)$, where $k$ ($\text{Å}^{-1}$) is the photoelectron wave vector. The contributions of the various Fe species to each sample were estimated using a linear combination fitting (LCF) of each $k^3$-weighted EXAFS spectrum with the spectra of the reference materials. LCF was conducted using no more than three minerals in a $k$ range of 2–9 $\text{Å}^{-1}$, by minimizing the residual of the fit, which was provided based on the residual value (the goodness-of-fit parameter, $R$) defined by the following:

$$R = \Sigma[k^3\chi_{\text{obs}}(k) - k^3\chi_{\text{cal}}(k)]^2 / \Sigma[k^3\chi_{\text{obs}}(k)]^2 \tag{1}$$

where $\chi_{\text{obs}}(k)$ and $\chi_{\text{cal}}(k)$ are the experimental and calculated absorption coefficients, respectively, at a given $k$. A smaller $R$ value indicates a better fit for the LCF procedure.

## 3. Results

The $k^3$-weighted Fe K-edge EXAFS spectra of ankerite and dolomite present similar characteristics, such as large positive amplitudes at around $k$ = 3.5, 4, 5.8, 7.5, and 9.5, whereas those of magnesite and siderite clearly differ (Figure 3A). The EXAFS spectra of S-bearing Fe minerals, including pyrite, pyrrhotite, jarosite, and $FeSO_4$, or Fe oxides present different spectra. The Fe K-edge EXAFS spectra also show clear distinctions among clay minerals such as chlorite, illite, kaolinite, nontronite, montmorillonite, and serpentine (Figure 3B). Although EXAFS spectra identify a group of clay minerals, the characteristics of spectra within the same group, such as nontronite (NAu-1, NAu-2, and NG-1) are similar and difficult to distinguish. Therefore, we used NAu-1, STX-1b, and low-temperature type serpentine as representatives of nontronite, montmorillonite, and serpentine, respectively, for the LCF of the ALH samples.

Almost all the measured EXAFS spectra of ALH samples show similar features, such as a large positive amplitude at around $k$ = 3.5, 5.8, and 7.7 (Figure 3C–F). The spectra, particularly within the range of $k$ = 3–5, are similar to those of magnesite in the reference material. Indeed, the EXAFS-LCF suggests a contribution from magnesite for all analytical spots (Table 3). The contributions of siderite and nontronite (NAu-1) are suggested for 29 and 18 analytical spots, respectively, from both carbonate and silica glass. Other species suggested by the EXAFS-LCF are magnetite, illite-smectite mixed layer (ISCz-1), pyrite, hematite, montmorillonite (STX-1b), and serpentine for 6, 4, 2, 1, 1, and 1 spots, respectively (Table 3).

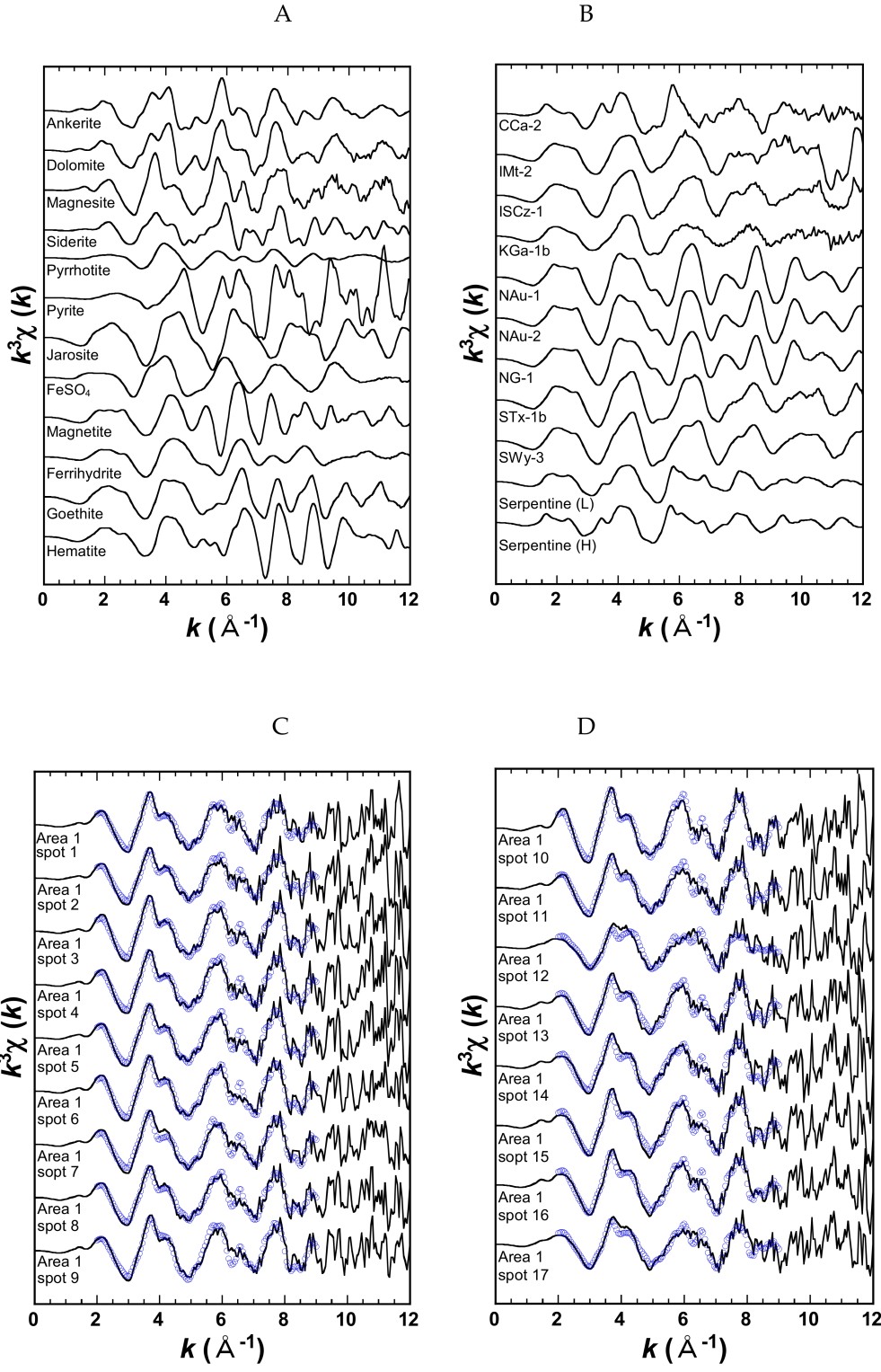

**Figure 3.** *Cont.*

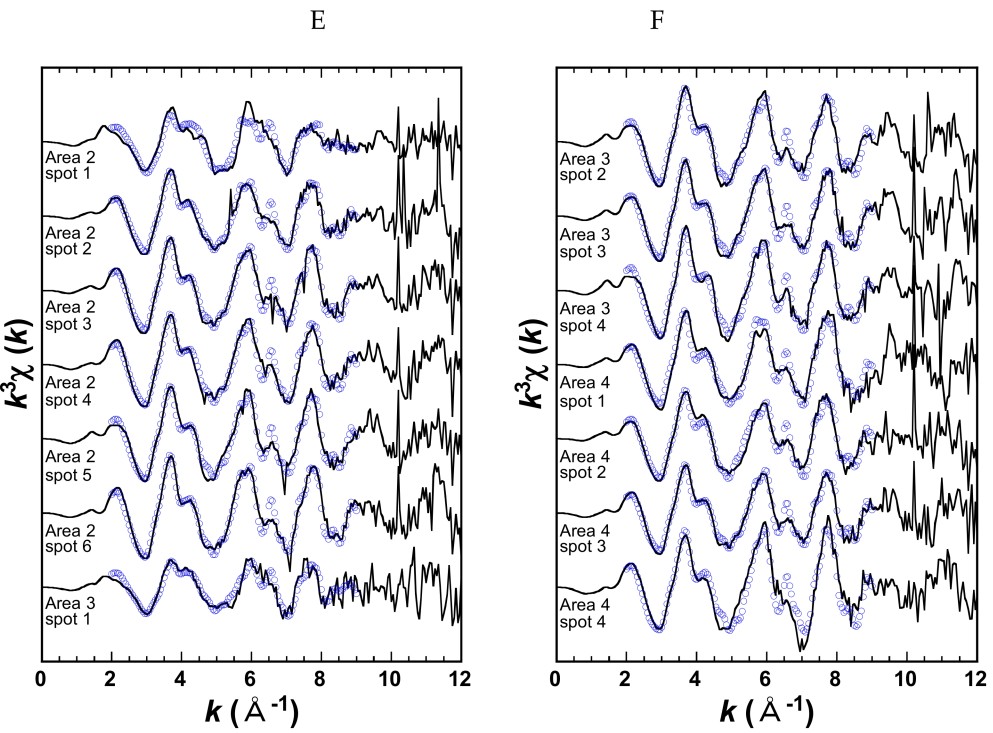

**Figure 3.** Iron K-edge EXAFS spectra of (**A**,**B**) standard minerals used to fit the measured ALH carbonate samples, and (**C**–**F**) μ-EXAFS analytical spots with fitting results. The solid black line denotes the measured spectra, and the blue circle shows the results of the fit using the parameters listed in Table 2. The detailed data are available in Supplementary Material Data S1.

**Table 3.** The linear combination fitting (LCF) result of the Fe K-edge μ-EXAFS analysis.

| Analytical Spots | | Mgs | Sd | STx | NAu | ISCz | Srp | Hem | Mag | Py | R (%) |
|---|---|---|---|---|---|---|---|---|---|---|---|
| *Area 1* | spot 1 | 59 | 24 | | 17 | | | | | | 0.122 |
| | spot 2 | 48 | 36 | 16 | | | | | | | 0.195 |
| | spot 3 | 48 | 39 | | | 13 | | | | | 0.116 |
| | spot 4 | 47 | 43 | | | 10 | | | | | 0.130 |
| | spot 5 | 42 | 49 | | | | | | 10 | | 0.117 |
| | spot 6 | 44 | 45 | | | | | | 11 | | 0.093 |
| | spot 7 | 63 | 26 | | | | | 11 | | | 0.122 |
| | spot 8 | 45 | 41 | | 13 | | | | | | 0.114 |
| | spot 9 | 45 | 39 | | 15 | | | | | | 0.089 |
| | spot 10 | 44 | 44 | | 12 | | | | | | 0.078 |
| | spot 11 | 46 | 37 | | 17 | | | | | | 0.155 |
| | spot 12 | 49 | | | | 38 | | | 14 | | 0.239 |
| | spot 13 | 41 | 36 | | | 23 | | | | | 0.142 |
| | spot 14 | 41 | 42 | | | | | | 16 | | 0.191 |
| | spot 15 | 49 | 33 | | 18 | | | | | | 0.107 |
| | spot 16 | 50 | 28 | | 22 | | | | | | 0.104 |
| | spot 17 | 45 | 26 | | 29 | | | | | | 0.217 |

**Table 3.** *Cont.*

| Analytical Spots | | Mgs | Sd | STx | NAu | ISCz | Srp | Hem | Mag | Py | *R* (%) |
|---|---|---|---|---|---|---|---|---|---|---|---|
| *Area 2* | spot 1 | 51 | | | 23 | | 27 | | | | 0.189 |
| | spot 2 | 56 | 31 | | 14 | | | | | | 0.078 |
| | spot 3 | 47 | 46 | | 7 | | | | | | 0.068 |
| | spot 4 | 46 | 41 | | 13 | | | | | | 0.064 |
| | spot 5 | 49 | 43 | | 8 | | | | | | 0.074 |
| | spot 6 | 52 | 40 | | 8 | | | | | | 0.046 |
| *Area 3* | spot 1 | 36 | 33 | | 31 | | | | | | 0.196 |
| | spot 2 | 43 | 49 | | | | | | 8 | | 0.065 |
| | spot 3 | 47 | 47 | | 6 | | | | | | 0.047 |
| | spot 4 | 52 | 39 | | 10 | | | | | | 0.060 |
| *Area 4* | spot 1 | 58 | 31 | | | | | | | 11 | 0.096 |
| | spot 2 | 43 | 42 | | | | | | 15 | | 0.077 |
| | spot 3 | 45 | 48 | | 7 | | | | | | 0.045 |
| | spot 4 | 37 | 52 | | | | | | | 11 | 0.127 |

Mgs: magnesite; Sd: siderite; STx: montmorillonite; NAu: nontronite; ISCz: illite-smectite mixed layer; Srp: serpentine; Hem: hematite; Mag: magnetite; Py: pyrite. Note that R (%) is calculated using Equation (1).

## 4. Discussion

### 4.1. Carbonate Species

The EXAFS-LCF does not suggest a contribution from ankerite or dolomite; however, it shows contributions from magnesite and siderite. In the chemical formula, ankerite $[CaFe(CO_3)_2]$ and dolomite $[CaMg(CO_3)_2]$ are located at the intermediate of the calcite $(CaCO_3)$–siderite $(FeCO_3)$ and calcite–magnesite $(MgCO_3)$ end members of the ternary diagram of carbonate, respectively. However, there are significant differences in their mineralogy, or crystallography. Both ankerite and dolomite are classified as members of dolomite groups, whereas magnesite and siderite are members of calcite groups. Although the crystal systems of both the dolomite and calcite groups are trigonal, the space groups of the dolomite and calcite groups are $R\bar{3}$ and $R\bar{3}c$, respectively. Although several studies have been conducted on the carbonate in ALH 84001, most studies in this area have described carbonate based on its chemical composition [16,45–47]; the mineralogy or crystallography of the carbonate has been poorly documented. In addition, a study using micro-Raman spectroscopy did not identify the detailed mineralogy of the carbonate [48]. Although XAFS analysis is a type of "fingerprinting method", the µ-EXAFS spectra of ALH carbonate clearly show a similarity to that of magnesite (Figure 3). Our data indicate that the carbonate minerals in ALH 84001 are dominated by the calcite group, mainly the magnesite-siderite solid solution.

### 4.2. Other Fe-Bearing Minerals

The EXAFS-LCF of some analytical points first suggested the contribution of Fe sulfate $(FeSO_4)$, possibly because the EXAFS oscillation of Fe sulfate shows an amplitude similar to that of magnetite (Figure 3A). However, we excluded the contribution from Fe sulfate and conducted an LCF again without adding $FeSO_4$ as a candidate. A previous study that conducted an S K-edge X-ray absorption near-edge structure (XANES) analysis showed that the S species in the ALH carbonate is carbonate-associated sulfate (CAS), presenting a different XANES spectrum to that of $FeSO_4$ [49]. The S XANES study also showed the contribution of sulfide mineral in six analytical spots of a total of 30 measurements. In our Fe analysis, pyrite was suggested from two analytical spots of slab carbonate with a contribution of ~10% to the Fe species (Table 3). Considering the Fe abundance in the carbonate, the S species should be affected by pyrite to a higher degree. This inconsistency can be attributed to the difference in the analytical area between the S and Fe speciation analyses. A previous TEM observation showed the presence of microcrystals of Fe sulfides [25]. The S K-edge analysis was conducted using an incident X-ray beam of 15 µm (vertical) × 15 µm (horizontal), whereas our Fe EXAFS analysis was conducted

using an X-ray beam size of 1–2 μm for both the vertical and horizontal dimensions. Therefore, such micro-sulfide crystals can be identified in our measurements, although the contribution of sulfide grains is diluted by the surrounding CAS using a much wider X-ray beam (S analysis).

The presence of magnetite has also been well documented [23,24,50], and our EXAFS-LCF suggested the contribution of magnetite from both rosette and slab carbonates. The magnetite crystals observed from ALH carbonate are also small, i.e., smaller than 100 nm, which is more than 10-times smaller than the incident X-ray beam in our analysis. However, similar to the sulfide grain, the Fe content in magnetite is much higher than that in the surrounding carbonate, which contributes to the Fe species. The contribution of hematite at the rosette core is consistent with previously reported Raman studies [48,51]. In our EXAFS-LCF, the contributions of several phyllosilicates, including nontronite, montmorillonite, illite-smectite mixed layer, and serpentine, are suggested (Table 3). Detailed SEM and TEM analyses showed the presence of smectite-like clays [14]. Among the clay minerals suggested by our EXAFS-LCF, nontronite, montmorillonite, and illite-smectite mixed layers are classified as the smectite group, and thus the suggestion of these minerals by our EXAFS-LCF supports the previous observations. However, the previous study reported the presence of smectite-like clay at (i) carbonate-pyroxene boundaries and (ii) a pyroxene surface not in contact with carbonate [14]. Meanwhile, our μ-EXAFS analyses suggest the contribution of smectite group clay from all analyzed phases, including inside the carbonate. Although the presence of smectite group clay (saponite) was also reported from the olivine fractures of other Martian meteorites [34], clays have not been reported from inside the carbonate. The suggestion of clay minerals from carbonate can account for the high $H_2O$ content in the carbonate [26]. A comparison between the two types of carbonates, rosette and slab, shows that rosette carbonate contains various types of Fe-bearing species other than carbonate, except for pyrite (Figure 4). Notably, our EXAFS-LCF suggests possible contributions from smectite groups such as Fe-Mg smectites (e.g., nontronite) and Al smectite (e.g., montmorillonite), whereas chlorite, which is widely observed on the Martian surface [11], is not suggested from ALH carbonate. It is also interesting that the serpentine is only suggested from silica glass. These differences reflect the evolution of the aqueous environment wherein the ALH carbonate (and silica glass) was formed.

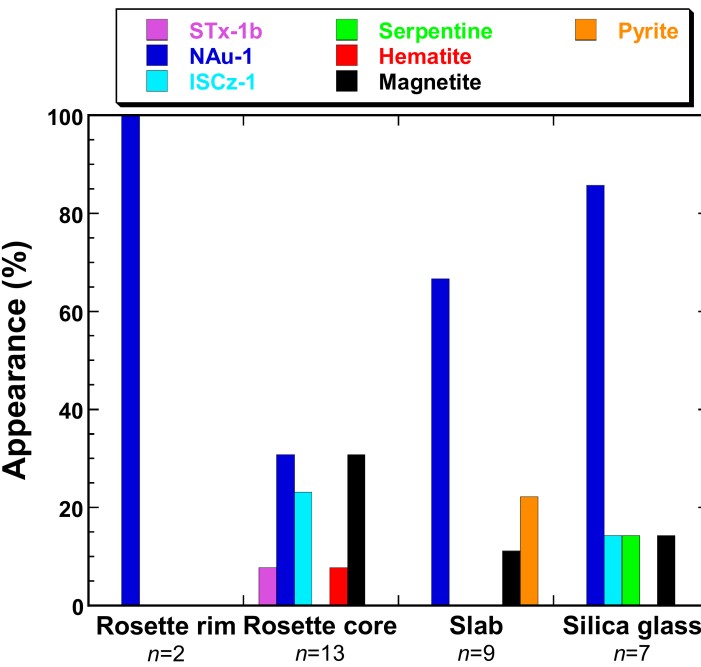

**Figure 4.** Fe-bearing minerals contributing to the analytical spots except for the carbonates (magnesite and siderite).

*4.3. Aqueous Environment of Ancient Mars*

The Fe-bearing species suggested by our EXAFS-LCF are formed (i) through coprecipitation with carbonate, or (ii) allochthonous and trapped in carbonate. One-dimensional transport thermochemical modeling indicates that nontronite (representing the smectite group) is formed under high water-rock ratio (W/R) conditions and coprecipitated with carbonate [29]. Meanwhile, the formation of chlorite occurs with the evolution of water under low W/R conditions where the amount of carbonate precipitation decreases. Thus, the absence of chlorite in our analysis is consistent with the model calculation. This calculation is also consistent with our results that the presence of the smectite group clay in the ALH carbonate results from coprecipitation with carbonate.

Although serpentine is generally formed with an alteration of olivine, which is scarcely observed in ALH 84001, the contribution of serpentine is suggested from silica glass (Figure 4). The thermochemical model suggests that the composition of chlorite is dominated by the Al-free endmember with a formula similar to serpentine and talc [29]. A small supply of $Fe^{2+}$ ions may result in the precipitation of serpentine in the evolved fluid. Alternatively, serpentine would be formed in an allochthonous manner through the alteration of olivine and trapped in the silica glass. Olivine is a common rock-forming mineral that is frequently observed in other Martian meteorites, and serpentine is widely observed on the Martian surface [11] and also suggested from nakhlite [34]. Considering that silica glass is the secondary phase formed after the formation of the ALH carbonate with different fluids, the allochthonous origin of the serpentine may be highly possible. In either case, the formation of serpentine requires a low W/R, suggesting a decrease in the volume of water, at least locally [29].

Our EXAFS-LCF does not suggest a contribution from goethite or ferrihydrite; however, magnetite and hematite are suggested. The possible coprecipitation of magnetite and carbonate are unlikely, because the precipitation of magnetite requires high pH and low W/R conditions, where the precipitation of carbonate is mainly terminated [29]. This is also supported by the TEM observation showing the allochthonous origin of magnetite, which is unrelated to any shock or thermal processing of the carbonates [25]. The discussion on the origin of magnetite is also applicable to hematite in the carbonate, as previous studies also suggested that the origin of hematite is the thermal dehydration of goethite or impact denaturation of siderite [51]. If hematite is also of allochthonous origin and was added to the carbonate from an outside source, this indicates that the Martian surface at ~4 Ga was already oxic, at least locally. The oxic Martian environment at ~4 Ga is consistent with the dominant contribution from CAS on S species, indicating the dominance of sulfate ions in the fluid from which ALH carbonate was precipitated [49]. In contrast, such an oxic environment may not be consistent with the recent discovery of N-bearing organics in the ALH carbonate, which requires less oxidizing conditions [38]. Therefore, the ancient aqueous environment on Mars was likely patchy with regard to the redox condition, at least in areas where the fluid resulted in the formation of the ALH carbonate. Alternatively, an extremely small window of Eh-pH conditions may explain the presence of either oxic (such as sulfate and hematite) or reducing (N-bearing organics) species.

## 5. Conclusions

In this study, an attempt was made to determine the Fe species in an ancient Martian meteorite using a microscale EXAFS analysis. The EXAFS analysis showed that Fe species in all ALH carbonates contained magnesite and siderite, whereas the presence of ankerite or dolomite was not suggested. The incorporation of smectite group clays, such as nontronite, montmorillonite, and illite-smectite mixed layers has been suggested for both rosette and slab carbonate. Magnetite was suggested for all analyzed phases, namely, rosette carbonate, slab carbonate, and silica glass, whereas hematite and pyrite were only suggested for rosette and slab carbonate, respectively.

The μ-EXAFS analyses support the results of a previous thermochemical modeling of the possible coprecipitation of carbonate with nontronite. The contribution of chlorite was

not suggested for ALH carbonate ($n = 24$) or silica glass ($n = 7$), which is also consistent with the model calculation. The presence of serpentine in silica glass indicates a decrease in the W/R, at least locally, after the formation of ALH carbonate. If the hematite contained in the rosette carbonate is of allochthonous origin, this indicates that the ancient aqueous environment of Mars was patchy with regard to the redox conditions. Such an oxic mineral was formed at a specific location, whereas N-bearing organics existed under more reducing conditions. A patchy redox condition, at least in areas where the fluid resulted in the formation of ALH carbonate, or an extremely small window of Eh-pH conditions, may explain the presence of both oxic and reducing species.

**Supplementary Materials:** The following are available online at https://www.mdpi.com/2075-163 X/11/2/176/s1, Data S1: The data presented in this study.

**Author Contributions:** Conceptualization, R.N., G.T., and T.U.; Measurement, R.N., G.T., I.K., T.U., and M.S.; writing—original draft preparation, R.N., and G.T.; writing—review and editing, T.U. and T.Y.; Funding acquisition, R.N. and T.U. All authors have read and agreed to the published version of the manuscript.

**Funding:** This study was partly supported by JSPS KAKENHI (Grant Numbers 15KK0153, 16H04073, 17H06458, 19H02007, and 19H01960). The speciation of Fe was measured with the approval of SPring-8 (Proposal Nos. 2017B1060, 2017B1854, 2018A1108, and 2018A1759) and KEK-PF (Proposal Nos. 2015G137, 2016G114, and 2017G116), respectively.

**Institutional Review Board Statement:** Not applicable.

**Informed Consent Statement:** Not applicable.

**Data Availability Statement:** Not applicable.

**Acknowledgments:** The authors are grateful to Ikuo Katayama of Hiroshima Univ. for providing the serpentine powders used in this study.

**Conflicts of Interest:** The authors declare no conflict of interest.

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
