# Peer review of "EXAFS Determination of Clay Minerals in Martian Meteorite Allan Hills 84001 and Its Implication for the Noachian Aqueous Environment"

_minerals, doi:10.3390/min11020176_

Round 1

Reviewer 1 Report

To the authors

This interesting manuscript is well-written, well-organized and gives a significant contribution to the knowledge of the aqueous environment of Mars.

On the whole, the paper surely deserves to be published in Minerals journal.

Nevertheless, some issues (see below) have to be addressed before publication. 

In particular, since I (and probably other researchers) did not know the possibility to successfully apply the method of the linear combination fitting (LCF) to EXAFS spectra, I would like to see in the text some references concerning this application.

Detailed comments and suggestion:

  • line 5: delete "and" in the middle of the list of authors
  • line 66: delete "such as charts" (NdR probably you ment such as cherts but this term is inappropriate here, see below comment for line 439)
  • line 67: insert "beam" (X-ray beam)
  • lines 90-91: there is a mistake in the Fig. 2 (see below my comment about this figure)
  • lines 104-105: ... siderite, and sulphides such as pyrite, ...
  • lines 175-176: this sentence looks contradictory to what is reported in the first sentence of the paragraph (lines 171-172), in the abstract and in the results; probably it is a typo and should be deleted.
  • line 178: insert "members of" (classified as members of).
  • line 178: insert "members of" (are members of).
  • lines 246-248: is this mechanism already known in literature? If so, it should be supported with properly references from literature.
  • line 283: analyzed instead of analytical.
  • line 392: the number 46 is still highlighted.
  • line 413 (caption of Figure 1): the caption reports the sentence "Boxes identify XRF mapping areas shown in Figure 2" but no boxes appear in the figure. You have to add the boxes or, otherwise, to delete the sentence.
  • line 429 (caption of Figure 2): there must be a mistake because images C, D,E,F, are absolutely identical!
  • line 439 (caption of Figure 3): the use of the word "chert" is totally inappropriate. Chert is a sedimentary rock composed of microcrystalline or cryptocrystalline quartz.

Author Response

line 5: delete "and" in the middle of the list of authors

line 66: delete "such as charts" (NdR probably you ment such as cherts but this term is inappropriate here, see below comment for line 439)

line 67: insert "beam" (X-ray beam)

Response: In accordance with these comments, we have rephrased the words and sentences as suggested.

lines 90-91: there is a mistake in the Fig. 2 (see below my comment about this figure)

line 429 (caption of Figure 2): there must be a mistake because images C, D,E,F, are absolutely identical!

Response: We apologize for the mistake. The correct figures were provided in the high resolution pdf file during initial submission, but the first and the corresponding author made mistakes while coping the figures to the word file. The mistake is corrected in the revised version.

lines 104-105: ... siderite, and sulphides such as pyrite, ...

Response: The sentence was rephrased as suggested.

lines 175-176: this sentence looks contradictory to what is reported in the first sentence of the paragraph (lines 171-172), in the abstract and in the results; probably it is a typo and should be deleted.

Response: This sentence was deleted to avoid confusion.

line 178: insert "members of" (classified as members of).

line 178: insert "members of" (are members of).

Response: In accordance with these comments, we have rephrased the words.

lines 246-248: is this mechanism already known in literature? If so, it should be supported with properly references from literature.

Response: This sentence was revised as follows:

Line 251: The thermochemical model suggests that the composition of chlorite is dominated by the Al-free endmember with a formula similar to the serpentine and talc [29].

Reference [29]: Melwani Daswani et al., Meteorit. Planet. Sci. 2016, 51, 2154–2174.

line 283: analyzed instead of analytical.

Response: In accordance with these comments, we have rephrased the word.

line 392: the number 46 is still highlighted.

Response: The reference number 46 is highlighted probably during the editorial process, because our original manuscript submitted to Minerals does not contain highlighted sentence.

line 413 (caption of Figure 1): the caption reports the sentence "Boxes identify XRF mapping areas shown in Figure 2" but no boxes appear in the figure. You have to add the boxes or, otherwise, to delete the sentence.

Response: We apologize for the inconvenience. The Figure with boxes identifying XRF mapping areas shown in Figure 2 is provided in the pdf version, but the boxes were disappeared in the word version. The boxes are added in Figure 1 in this revision.

line 439 (caption of Figure 3): the use of the word "chert" is totally inappropriate. Chert is a sedimentary rock composed of microcrystalline or cryptocrystalline quartz.

Response: The sentence was rephrased as follows:

Line 448: Iron K-edge EXAFS spectra of (A)–(B) standard minerals used to fit the measured chert ALH carbonate samples, and (C)–(F) μ-EXAFS analytical spots with fitting results.

Reviewer 2 Report

Review of: EXAFS Determination of Clay Minerals in Martian Meteorite Allan Hills 84001 and Its Implication for The Noachian Aqueous  Environment

By Nakada et al.

Line 40.  The radiometric age that this refers to is K-Ar on detrital minerals. The smectite in contrast is almost certainly diagenetic, and much younger than this ie early Hesperian (Vaniman et al. 2014, Leveille et al. 2014 etc.)

Line 44 and Elsewhere.  Although the conclusion of this manuscript is that magnesite-siderite is dominant, the actual composition of ALH84001 carbonates (e.g. Harvey and McSween 1995) is Ca-rich with a large calcite component.  This needs to be described more clearly. A better comparison to known marina carbonate could have been given e.g. see Bridges J.C., Hicks L.J, Treiman A. (2019) Carbonates on Mars. In ‘Volatiles on Mars’.  1st edition. Elsevier. 

Lin 144.  It looks like some standards have been used e.g. ‘we used NAu-1, STX-1b, and low-tempera- ture type serpentine as representatives of nontronite, montmorillonite, and serpentine’ but more description and of what these minerals are (e.g. composition, locality ec) together with their EXAFS spectra would help, including Table 2.

respectively, for the LCF of the ALH samples.

Line 189, Lin 250 and Elsewhere. In the descriptions of martian phyllosilicates mention should be made of those in the nakhlite martian meteorites. They contain ferric saponite in olivine fractures and a 1:1 serpentine-like mineral in mesostasis e.g. (Hicks et al. GCA 2014).  

Line 255. The authors say ‘ formation of serpentine requires a low W/R,’ – this assertion needs some references and explanation.

Lin2 292. This final sentence needs rewriting, its not clear what geological scenario are describing, a series of different fluid events at the same location on Mars?

Figure 1. Needs a scale bar. Its also not a very good image, can a better one be swapped ion for it?

Table 1, 2.  What about adding the clay standards? 

Author Response

Line 40.  The radiometric age that this refers to is K-Ar on detrital minerals. The smectite in contrast is almost certainly diagenetic, and much younger than this ie early Hesperian (Vaniman et al. 2014, Leveille et al. 2014 etc.).

Response: In accordance with the comment, the sentence was rephrased as follows:

Line 39: Smectites were also observed in ~4.21 Ga old crater-rim-derived rocks in the Gale crater, possibly formed by diagenesis in the early Hesperian [12, 13].

Reference [12]: Farley et al., Science 2014, 343, 1247166.

Reference [13]: Vaniman et al., Science 2014, 343, 1243480.

Line 44 and Elsewhere.  Although the conclusion of this manuscript is that magnesite-siderite is dominant, the actual composition of ALH84001 carbonates (e.g. Harvey and McSween 1995) is Ca-rich with a large calcite component.  This needs to be described more clearly. A better comparison to known marina carbonate could have been given e.g. see Bridges J.C., Hicks L.J, Treiman A. (2019) Carbonates on Mars. In ‘Volatiles on Mars’.  1st edition. Elsevier.

Response: As pointed by the comment, our speciation analysis only focused on the speciation of Fe in ALH carbonate. The lack of an adequate reference of calcite containing Fe (calcite-siderite solid solution) resulted in the conclusion of the dominance of magnetite-siderite. In accordance with the comment, we clearly describe that (i) Fe speciation analysis showed that the dominance of members of calcite group carbonate through the manuscript, and (ii) the lack of adequate reference of calcite-siderite solid solution in the section 2.2.1. (Reference materials).

Line 144.  It looks like some standards have been used e.g. ‘we used NAu-1, STX-1b, and low-temperature type serpentine as representatives of nontronite, montmorillonite, and serpentine’ but more description and of what these minerals are (e.g. composition, locality etc) together with their EXAFS spectra would help, including Table 2. respectively, for the LCF of the ALH samples.

Table 1, 2.  What about adding the clay standards?

Response: In accordance with these comments, we added the descriptions of reference clay minerals as a new Table (Table 2). By this addition, the number of Table 2 were changed as Table 3. The EXAFS spectra of clay minerals are already shown in Figure 3B.

Line 189, Line 250 and Elsewhere. In the descriptions of martian phyllosilicates mention should be made of those in the nakhlite martian meteorites. They contain ferric saponite in olivine fractures and a 1:1 serpentine-like mineral in mesostasis e.g. (Hicks et al. GCA 2014). 

Response: In accordance with the comment, the following sentences were added and rephrased:

Line 227: Although the presence of smectite group clay (saponite) has also reported from olivine fracture of other Martian meteorites [34], clays have not been reported from inside the carbonate.

Line 255: Olivine is a common rock-forming mineral that is frequently observed in other Martian meteorites, and serpentine is widely observed on the Martian surface [11] and also suggested from nakhlite [34].

Reference [11]: Ehlmann et al., Nature 2011, 479, 53–60.

Reference [34]: Hicks et al. Geochim. Cosmochim. Acta 2014, 136, 194–210.

Line 255. The authors say ‘ formation of serpentine requires a low W/R,’ – this assertion needs some references and explanation.

Response: This sentence was revised as follows:

Line 260: In either case, the formation of serpentine requires a low W/R, suggesting a decrease in the volume of water, at least locally [29].

Reference [29]: Melwani Daswani et al., Meteorit. Planet. Sci. 2016, 51, 2154–2174.

Line 292. This final sentence needs rewriting, its not clear what geological scenario are describing, a series of different fluid events at the same location on Mars?

Response: In accordance with the comment, the following sentence was added:

Line 298: A patchy redox condition at least in areas where the fluid resulted in the formation of ALH carbonate, or an extremely small window of Eh-pH conditions may explain the presence of either oxic and reducing species.

Figure 1. Needs a scale bar. Its also not a very good image, can a better one be swapped ion for it?

Response: In accordance with the comment, a scale bar was added in the Figure 1. Regarding the image, a high-resolution version is uploaded in the pdf version.